# REVIVE: Regional Visual Representation Matters in Knowledge-Based Visual Question Answering

**Yuanze Lin**♣*  **Yujia Xie**♠  **Dongdong Chen**♠
**Yichong Xu**♠  **Chenguang Zhu**♠  **Lu Yuan**♠
♣ University of Washington  ♠ Microsoft
yuanze@uw.edu  {yujiaxie, dochen, yicxu}@microsoft.com

## Abstract

This paper revisits visual representation in knowledge-based visual question answering (VQA) and demonstrates that using regional information in a better way can significantly improve the performance. While visual representation is extensively studied in traditional VQA, it is under-explored in knowledge-based VQA even though these two tasks share the common spirit, i.e., rely on visual input to answer the question. Specifically, we observe that in most state-of-the-art knowledge-based VQA methods: 1) visual features are extracted either from the whole image or in a sliding window manner for retrieving knowledge, and the important relationship within/among object regions is neglected; 2) visual features are not well utilized in the final answering model, which is counter-intuitive to some extent. Based on these observations, we propose a new knowledge-based VQA method ***REVIVE***, which tries to utilize the explicit information of object regions not only in the knowledge retrieval stage but also in the answering model. The key motivation is that object regions and inherent relationship are important for knowledge-based VQA. We perform extensive experiments on the standard OK-VQA dataset and achieve new state-of-the-art performance, *i.e.*, **58.0%** accuracy, surpassing previous state-of-the-art method by a large margin (**+3.6%**). We also conduct detailed analysis and show the necessity of regional information in different framework components for knowledge-based VQA. Code is publicly available at https://github.com/yzleroy/REVIVE.

## 1 Introduction

Many vision-based decision making processes in our daily life go beyond perception and recognition. For example, if we see a salad bowl in the deli bar, our decision on whether to buy it does not only depend on what is in the bowl, but also the calories in each of the item. This motivates the knowledge-based Visual Question Answering (VQA) task [22], which extends traditional VQA task [2] to solve more complex problems, *i.e.*, where commonsense knowledge is required to answer the open-domain questions.

By definition, knowledge-based VQA takes three different information sources to predict the answer: input visual information (image), input question, and the external knowledge. While existing research on knowledge-based VQA mainly focuses on improving the incorporation of external knowledge, this paper focuses on improving the object-centric visual representation and presents a comprehensive empirical study to demonstrate that visual features matter in this task.

Intuitively, visual information should be well used for both knowledge retrieval and final answering. However, we find existing state-of-the-art (SOTA) methods [37, 8] in such domain have not fully

---

*Work done during an internship at Microsoft Redmond.

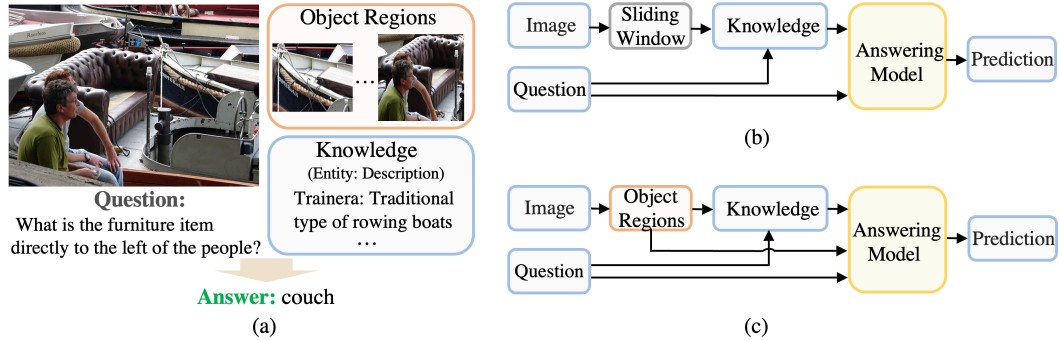

Figure 1: (a) An example from OK-VQA dataset, our method utilizes the retrieved knowledge and object-centric regions to solve the question. (b) The pipeline of previous state-of-the-art method KAT [8]. (c) The pipeline of our proposed *REVIVE*.

utilized it. On the one hand, they simply use either the whole image or a sliding window on the image to retrieve the external knowledge. On the other hand, they ignore the essential visual information (*i.e.*, object-centric representations) in the final answering model. In other words, they fuse only the retrieved knowledge and the question as a pure natural language processing (NLP) model to obtain the answer, a typical method [8] is illustrated in Figure 1 (b).

In this paper, we revisit visual representation in knowledge-based VQA, and argue that the information of object regions and their relationship should be considered and used in a dedicated way. The underlying motivation is shown in Figure 1 (a), which demonstrates that understanding the objects and their relationship is necessary. To this end, we propose **REVIVE** to better utilize **RE**gional **VI**sual Representation for knowledge-based **V**isual qu**E**stion answering. It not only exploits the detailed regional information for better knowledge retrieval, but also fuses the regional visual representation into the final answering model. Specifically, we first use the object detector GLIP [16] to locate the objects, and then use the cropped region proposals to retrieve different types of external knowledge. Finally, we integrate the knowledge together with the regional visual features into a unified transformer based answering model for final answer generation.

We perform extensive experiments on the OK-VQA dataset [22], and the proposed *REVIVE* achieves the SOTA performance of **58.0%** accuracy, a **3.6%** absolute improvement from the results of previous SOTA method [8].

We summarize our contribution as follows:

(a) We systematically explore how to better exploit the visual feature to retrieve knowledge. The empirical results suggest the region-based approach performs the best, compared to whole image-based and sliding window-based approaches.

(b) We integrate the regional visual representation, retrieved external and implicit knowledge into a transformer-based question answering model, which can effectively leverage the three information sources for solving knowledge-based VQA.

(c) Our proposed REVIVE achieves the state-of-the-art performance on OK-VQA dataset, *i.e.*, **58.0%** accuracy, surpassing the previous methods by a large margin.

## 2 Related Work

**Knowledge-Based VQA.** Knowledge-based VQA [22] aims to predict answers for general questions by leveraging external knowledge beyond image content. Early works [34, 33] introduce external knowledge to solve visual question answering (VQA) tasks. OK-VQA dataset [22] is the first large-scale dataset with questions that need be answered using external knowledge instead of a provided fixed knowledge base [33]. Recent studies [34, 33, 24, 23, 40, 36, 21, 37, 8] integrate different knowledge from various external knowledge resources, *e.g.*, ConceptNet [28], Wikipedia [31], *etc*, for solving knowledge-based VQA. Later, PICa [37] regards large language models, e.g., GPT-3 [3] as an implicit knowledge source and employs it [3] to get answer prediction based on textual prompts. Inspired by the recent success of knowledge-retrieved methods [11, 10] that leverage

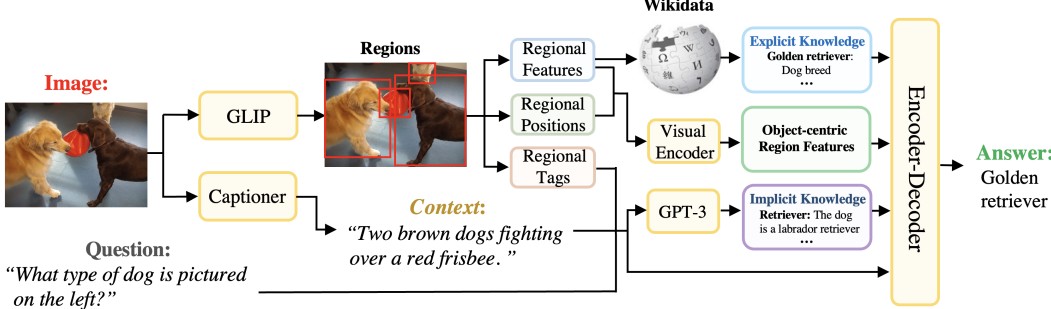

Figure 2: **The illustration of *REVIVE*.** It exploits regional information (*i.e.*, features, positions and tags), question and context to retrieve different types of knowledge. In addition, it also incorporates learned object-centric region features with retrieved knowledge for answer generation.

external knowledge retrieval with language generative models for open-domain question answering, KAT [8] exploits the FiD reader [11] to perform knowledge reasoning over retrieved implicit and explicit knowledge. Our work instead emphasizes revisiting the visual representation for knowledge retrieval, *i.e.*, resorting to regional visual representation. In addition, we propose to incorporate object-centric regional visual representation together with retrieved knowledge in the answer generative model. Several works [23, 6, 5, 27, 24] have incorporated visual embeddings or captions in predicting the final answers. However, these works target at different settings and they haven't fully explored how to better use regional representations to retrieve knowledge.

**Vision-Language Models.** Recent years have witnessed the rapid development of vision-language models [30, 12, 38, 29, 17, 39, 32, 35]. Those works usually first pre-train a neural network on a large-scale image-text dataset and then finetune the models for solving specific vision-language tasks. Among them, VinVL [39] aims to learn the object-centric representation. CLIP [25] pre-trains the models with large-scale text-image pairs by contrastive learning. GLIP [38] reformulates the pre-training process by unifying object detection and phrase grounding. Our method uses the three models as sub-modules to identify object-centric regions and retrieve knowledge for knowledge-based VQA task.

## 3  Proposed Method

Knowledge-based VQA task [22] seeks to answer questions based on external knowledge beyond images. Specifically, let us denote a knowledge-based VQA dataset as $\mathcal{D} = \{(I_i, Q_i, A_i)\}_{i=1}^{N}$, where $I_i$, $Q_i$ and $A_i$ denote the input image, question and answer of the $i$-th sample respectively, and $N$ is the number of total samples. Given the dataset, the goal is to train a model with parameter $\theta$ to generate the answer $A_i$ with input $I_i$ and $Q_i$.

In this section, we introduce our method *REVIVE*. Figure 2 shows an overview of the method. We leverage the detected regions of the input image to obtain the object-centric region features and retrieve explicit knowledge. Meanwhile, we prompt GPT-3 [3] by regional tags, question and context to retrieve implicit knowledge. After that, the regional visual features, retrieved knowledge, and the text prompt consists of regional tags, question and context will then be fused into a encoder-decoder module to generate the answer. We explain more details in Section 3.1, 3.2 and 3.3.

### 3.1  Regional Feature Extraction Module

Given an image $I$, we first adopt a object detector to give us the positions of region proposals,

$$\mathcal{B} = \{b_j\}_{j=1}^{M} = D(I), \tag{1}$$

where $\mathcal{B} = \{b_j\}_{j=1}^{M}$ is the set of bouding boxes, $M$ is the number of detected boxes, and $D(\cdot)$ is the object detector.

Here, we adopt $D(\cdot)$ as the visual grounding model GLIP [16]. We use the text prompt "`Detect: person, bicycle, car, ..., toothbrush`", which contains all object categories of MSCOCO

dataset [18]. In this way, the model can provide us with all bounding boxes associated with those categories.

After we get the bounding boxes $\mathcal{B}$ of interested objects from GLIP, We crop the image $I$ according to $\mathcal{B}$ to obtain region proposals $\mathcal{R} = \{r_j\}_{j=1}^{M}$. We then extract the object-centric visual features from the proposals: $v_j = E(r_j)$, where $v_j \in \mathbb{R}^S$ is the visual embedding of the $j$-th proposal, $S$ is the embedding dimension and $E(\cdot)$ represents the image encoder. Inspired by the strong transferring capability of recent contrastively trained vision-language models, we adopt the visual encoder of CLIP [25] as our image encoder $E(\cdot)$. We use the encoding of [CLS] token as the final embedding.

To understand the relationship between/among the objects, we find it also important to introduce the position information $\mathcal{B}$ along with its regional visual features.

In addition to the embeddings, explicitly obtaining the description of each region proposal in the textual format is also helpful for knowledge retrieval. For the contrastively trained vision-language models, the training loss explicitly encourages inner product between the image embedding and the text embedding to be larger if the image and the text are well-aligned. Therefore, such a model is capable of selecting the tags that describe the image from a set of customized tags $\bar{\mathcal{T}}$ by computing the inner product. Denote the language encoder of CLIP as $T(\cdot)$. Given a set of tags $\bar{\mathcal{T}} = \{t_i\}_{i=1}^{N_1}$, $N_1$ is the number of total tags, we compute the inner product between the region proposals and all tags, and adopt the tags with the top-$P$ similarities as the description of the region proposals,

$$\mathcal{H} = \{h_p\}_{p=1}^{P} = \arg \operatorname*{TopP}_{t_i \in \bar{\mathcal{T}}} \langle E(r_j), T(t_i) \rangle, \quad j = 1, \cdots, M, \tag{2}$$

where $\langle \cdot, \cdot \rangle$ is the inner product, $P$ denotes the number of the obtained regional tags and $\mathcal{H}$ means the retrieved regional tags.

In complement to the localized textual description $\mathcal{H}$, we adopt a caption model to explicitly describe the relationships between the major objects and provide more context,

$$c = C(I), \tag{3}$$

where $C(\cdot)$ is the caption model. For example, in Figure 2, the context "Two brown dogs fighting over a red frisbee" provides us with the essential relationships between the objects, e.g., fighting *over* a red frisbee. Here, we adopt Vinvl [39] as the caption model $C(\cdot)$.

In summary, we extract regional visual and positional information as $\{v_j\}_{j=1}^{M}$ and $\{b_j\}_{j=1}^{M}$, and textual descriptions for the objects and the relationship between the objects as $\mathcal{H}$ and $c$. In the next section, we will elaborate on how we use these regional information sources to retrieve external knowledge.

### 3.2 Object-Centric Knowledge Retrieval Module

Inspired by KAT [8], we consider both the explicit knowledge and implicit knowledge. But different from it, we utilize regional visual information to help boost the final performance.

#### 3.2.1 Explicit Regional Knowledge

Since the questions from knowledge-based VQA [22] are general and open-ended, introducing external knowledge is important for model to generate accurate answers by providing extra and complementary knowledge beyond visual contents of input images.

**External Knowledge Base.**     We construct an external knowledge base $\mathcal{Q}$ by constructing a subset from Wikidata [31] following KAT [8]. Specifically, we extract 8 commonly appeared categories, *i.e.*, Role, Point of interest, Tool, Vehicle, Animal, Clothing, Company, Sport, to form the subset $\mathcal{Q}$. Each item in $\mathcal{Q}$ consists of an entity and a corresponding description, *e.g.*, one entity and its description can be "pegboard" and "board wall covering with regularly-spaced holes for insertion of pegs or hooks" respectively.

**Regional Knowledge Retrieval.**     As mentioned earlier, vision-language models like CLIP are capable of selecting the most relevant text from a set of texts. We reformat the entries in knowledge base $\mathcal{Q}$ as "{entity} is a {description}", and denote the reformatted text set as $\mathcal{T}$. We retrieve the

top-$K$ most relevant knowledge entries among *all the regional proposals* as explicit knowledge $\mathcal{E}$,

$$\mathcal{E} = \{e_k\}_{k=1}^K = \arg\operatorname*{TopK}_{d_i \in \mathcal{T}}\langle E(r_j), T(d_i)\rangle, \quad j = 1, \cdots, M, \tag{4}$$

where $K$ denotes the number of retrieved explicit knowledge samples. In our implementation, we use FAISS [13] to speed up the computation of Equation (2) and (4).

### 3.2.2 Implicit Knowledge with Regional Descriptions

Large language models, *e.g.*, GPT-3 [3], not only excel in many language tasks, but also memorize lots of commonsense knowledge from its training corpus [37]. Therefore, we exploit GPT-3 [3] as our implicit knowledge base by reformulating the task as open-domain question answering.

**Context-Aware Prompt with Regional Descriptions.** We design the textual prompt based on question $Q$, caption $c$, and *tags* $\mathcal{H}$. Different from PICa [37] and KAT [8] that use whole-image feature to get the tags, we utilize fine-grained regional features to extract regional tags. Specifically, we adopt the prompt $X$ to be "`context:` {caption} + {tags}. `question:` {question}". In this way, the language model is also supplemented with regional visual information.

**Implicit Knowledge Retrieval.** Finally, we query GPT-3 model [3] which takes the reformulated prompt $X$ as input, and obtain predictive answer. Since some of the questions may have ambiguity, we follow the prompt tuning procedure of PICa [37] and get answer candidates $\{o_u\}_{u=1}^U$. In addition to answer prediction, we also aim for acquiring corresponding explanation $e_u$ from GPT-3 model to obtain more context information. To be more specific, the corresponding explanation is acquired by feeding the text prompt "{question} {answer candidate}. `This is because`" into GPT-3. Note that "{question}" and "{answer candidate} are input question $Q$ and GPT-3's answer $o_u$ for image $I$ respectively. The final retrieved implicit knowledge can be denoted as $\mathcal{I} = \{(o_u, e_u)\}_{u=1}^U$.

### 3.3 Encoder-Decoder Module with Object-Centric Visual Features

Once we've retrieved the explicit and implicit knowledge and the regional information, we utilize the FiD network structure [11] to encode and decode retrieved knowledge and regional information.

**Knowledge Encoder.** For the explicit knowledge, we reformat the input text as "`entity:` {entity} `description:` {description}", where the entity and the description is from the entries in the retrieved explicit knowledge $\mathcal{E}$. We denote this text as $h_k$, where $k = 1, \cdots, K$.

For implicit knowledge, we adopt input format as "`candidate:` {answer} `evidence:` {explanation}", where answer is the retrieved answer $o_u$ and explanation is $e_u$. Here, $u = 1, \cdots, U$, where $U$ is the number of answers provided by GPT-3. We denote the input text as $s_u$.

We then encode the knowledge in textual format by the FiD's encoder [30], which is denoted as $F_e$,

$$\alpha_k = F_e(h_k), \quad \beta_u = F_e(s_u), \tag{5}$$

in which $\alpha_k \in \mathbb{R}^D$, $\beta_u \in \mathbb{R}^D$ and $D$ means the embedding dimension.

**Regional Visual Encoder.** We introduce a visual encoder for the regional visual embeddings $\{v_j\}_{j=1}^M$ and positional coordinates $\{b_j\}_{j=1}^M$. We feed $v_j$ and $b_j$ into two different fully connected layers, stack the outputs into a sequence of vectors, and then feed them into a transformer encoder $F_v$,

$$f = F_v(\operatorname{Concat}(\operatorname{FC}_1(v_1), \operatorname{FC}_2(b_1), \cdots, \operatorname{FC}_1(v_M), \operatorname{FC}_2(b_M))), \tag{6}$$

where $f \in \mathbb{R}^{(2M) \times D}$, $\operatorname{FC}_1(\cdot)$ and $\operatorname{FC}_2(\cdot)$ are two different fully-connected layers, $\operatorname{Concat}(\cdot)$ is the concatenation operation along a new dimension.

**Context-aware Question Encoder.** To better leverage the context information, we replace the input question $Q$ by the context-aware prompt $X$, we then encode it by the same transformer encoder $F_e(\cdot)$,

$$q = F_e(X), \tag{7}$$

where $q \in \mathbb{R}^D$ and $q$ means encoded context-aware question.

**Generative Decoder.** We have obtained the knowledge encoding $\{\alpha_k\}_{k=1}^K$ and $\{\beta_u\}_{u=1}^U$, visual encoding $f$, and context-aware question encoding $q$. Note that as the outputs of the encoder $F_e$, they

are all sequences of vectors. We then concatenate these vectors along the first dimension, and feed them into the FiD's decoder $F_d$,

$$y = F_d(\text{Concat}(\alpha_1, \cdots, \alpha_K, \beta_1, \cdots, \beta_U, f, q)), \tag{8}$$

where $y$ means the generated answer. The cross entropy loss function is adopted to train the model,

$$\mathcal{L} = -\sum_{\ell=1}^{L} \log p_\theta(\bar{y}_\ell | y_{<\ell}), \tag{9}$$

in which $L$ is the length of the ground truth answer text, $\bar{y}_\ell$ is ground truth text at the position $\ell$ and $\theta$ is the model parameters.

**Model Ensemble.** To generate more accurate answers, one promising method is to leverage multiple trained models, *i.e.*, model ensemble. In our experiments, we just train three models whose initialized seeds are different, and then the most frequent result among the generated results from these three models is selected as final answer prediction for each sample.

### 3.4 Relationship to Existing Works

Inspired by KAT [8], *REVIVE* also retrieves two types of knowledge, *i.e.*, implicit and explicit knowledge. Different from KAT, we explore how to better use visual features to retrieve knowledge. Motivated by the fact that the retrieved knowledge should also corresponds to individual concepts in the images in addition to the global theme, we use extracted regional features to retrieve external knowledge, and use regional descriptions to obtain the implicit knowledge. Moreover, we integrate the visual representation of object regions with retrieved knowledge in the answer generative model. The pipeline differences between KAT [8] and our method can be explained in Figure 1.

There're two works [36, 21] that leverage visual regions for knowledge-based VQA as well. However, MAVEx [36] considers object regions as a kind of knowledge without using their visual representation to retrieve other knowledge, KRISP [21] utilizes object regions to learn implicit knowledge by a transformer-based model and retrieve external knowledge by the text symbols of these regions, while our proposed *REVIVE* explores how to better leverage visual representation to retrieve knowledge and integrate their visual features with retrieved knowledge into the answering model.

## 4 Experiments

### 4.1 Experimental Setup

**Dataset.** OK-VQA dataset [22] is selected for evaluation, which is currently the largest knowledge-based VQA dataset. OK-VQA dataset includes 14055 questions associated with 14031 images from MSCOCO dataset [18]. Its questions cover a variety of knowledge categories, and are annotated by Amazon Mechanical Turkers. The training and testing split consist of 9009 and 5046 samples respectively. Each data sample is made up of one question, one corresponding image and 10 ground-truth answers. To construct the general domain tag set $\bar{\mathcal{T}}$, we collect the most frequently searched 400K queries in Bing Search as the tags.

**Pre-processing.** We utilize the pre-trained visual grounding model GLIP-T [16] to detect object-centric region proposals by using its default prompt "`Detect: person, bicycle, car, ..., toothbrush`", which contains all object categories of MS-COCO dataset [18]. The captions of images are obtained by the pre-trained Vinvl-Large model [39]. For explicit knowledge and regional tag retrieval, we choose CLIP model (ViT-B/16 variant) [25]. In our experiments, we adopt $U$, $K$, $M$ and $P$ as 5, 40, 36 and 30 respectively. Note that the models of CLIP, GLIP, Vinvl and GPT-3 are all frozen during usage.

**Implementation Details.** We use $4 \times$ NVIDIA V100 32Gb to train models for 10K steps, with a batch size of 8. The learning rate is $8\text{e}^{-5}$ and AdamW [19] is chosen as optimizer. The warm-up steps are 1K and the trained models are evaluated every 500 steps. We initialize our model with the pre-trained T5 model [26], *i.e.*, T5-large, following KAT [8]. The encoder $F_v$ in Equation (6) consists of 9 transformer layers [30]. Note that we evaluate the prediction results after normalization, and the normalization process mainly includes removing articles, punctuation and duplicated whitespace and lowercasing [4, 14].

**Evaluation Metric.** In our experiments, we choose the soft accuracy of VQAv2 [2] as evaluation metric for comparison.

Table 1: Results comparison with existing methods on OK-VQA dataset [22], the evaluation metric (*i.e.*, accuracy) is in %.

| Method | Knowledge Resources | Accuracy (%) |
|---|---|---|
| Q only [22] | - | 14.9 |
| MLP [22] | - | 20.7 |
| BAN [22] | - | 25.1 |
| BAN+AN [22] | Wikipedia | 25.6 |
| MUTAN [22] | - | 26.4 |
| BAN+KG-AUG [15] | Wikipedia+ConceptNet | 26.7 |
| MUTAN+AN [22] | Wikipedia | 27.8 |
| ConceptBERT [7] | ConceptNet | 33.7 |
| KRISP [21] | Wikipedia + ConceptNet | 38.4 |
| Visual Retriever-Reader [20] | Google Search | 39.2 |
| MAVEx [36] | Wikipedia+ConceptNet+Google Images | 39.4 |
| PICa-Base [37] | Frozen GPT-3 (175B) | 43.3 |
| PICa-Full [37] | Frozen GPT-3 (175B) | 48.0 |
| KAT (Single) [8] | Wikidata+Frozen GPT-3 (175B) | 53.1 |
| KAT (Ensemble) [8] | Wikidata+Frozen GPT-3 (175B) | 54.4 |
| REVIVE (Single) | Wikidata+Frozen GPT-3 (175B) | **56.6** |
| REVIVE (Ensemble) | Wikidata+Frozen GPT-3 (175B) | **58.0** |

## 4.2 Comparison with State-of-the-art Methods

As shown in Table 1, we can see that previous works (*e.g.*, KRISP [21], Visual Retriever-Reader [20] and MAVEx [36]) achieve similar performances, about 38.4% to 39.4% accuracy. Until recently, PICa [37] is the first one that exploits the pre-trained language model GPT-3 [3] as knowledge base for knowledge-based VQA task and KAT [8] further introduces Wikidata [31] as an external knowledge resource, these two works obtain significant performances compared with previous ones.

The proposed *REVIVE* can outperform all existing methods by large margins. Specifically, even using the same knowledge resources (*i.e.*, Wikidata [31] and GPT-3 [3]), our single model can achieve **56.6%** accuracy versus previous state-of-the-art method KAT's **53.1%** accuracy, when using model ensemble, our method can achieve **58.0%** accuracy compared with KAT's **54.4%** accuracy. These results demonstrate the effectiveness of the proposed approach.

## 4.3 Ablation Study

Next, we conduct extensive ablation studies on the single model to figure out the influence of each component of *REVIVE*.

**Effect of Region Proposal Number.** We perform the ablation study to figure out the effect of using different region proposal numbers. The results are displayed in Table 2. It can be observed that when the region proposal number is 36, the model achieves optimal performance. We conjecture that when the number of region proposals is too large, there are some meaningless and noisy region proposals, while if the number of region proposals is too small, many essential object-centric regions are ignored, which both hurt the model's performance.

**Different Knowledge Retrieval Methods.** The way of utilizing visual representation for retrieving knowledge plays an important role in knowledge-based VQA. We show the results of using three kinds of knowledge retrieval methods, *i.e.*, image-based, sliding window-based and region-based, in Table 5. Note that *sliding window-based* approach follows KAT [8]. Specifically, we first resizes input images to $384 \times 384$ and then crop the images with a sliding window whose size is $256 \times 256$ and stride size is 128. We can observe that the proposed *region-based* approach achieves best performance and surpasses *sliding window-based* method by **1.8%** points, which can validate the effectiveness of exploiting region-based visual representation for retrieving knowledge.

**Effect of Regional Tag Number.** In order to introduce more semantics into contexts, we propose to add region-aware descriptions (*i.e.*, regional tags) behind given contexts. We report the results of using different regional tag number for text prompt $X$ in Table 3. The results show that when the

Table 2: Ablation study on using different region proposal number.

| # of region proposals | Accuracy (%) |
|---|---|
| 5 | 54.7 |
| 18 | 55.8 |
| 36 | **56.6** |
| 50 | 56.2 |

Table 3: Ablation study on using different regional tag number.

| # of regional tags | Accuracy (%) |
|---|---|
| 8 | 56.2 |
| 24 | 56.4 |
| 30 | **56.6** |
| 50 | 56.3 |

Table 4: Ablation study on adopting bounding box coordinates.

| Positional coordinates | Accuracy (%) |
|---|---|
| ✗ | 55.8 |
| ✓ | **56.6** |

Table 5: Ablation study on adopting different methods for retrieving knowledge.

| Method | Accuracy (%) |
|---|---|
| Image-based | 53.2 |
| Sliding window-based | 54.8 |
| Region-based | **56.6** |

number of regional tags is 30, it achieves optimal performances. In fact, when the number of regional tags is too large, we'll retrieve relatively irrelevant object tags, sacrificing the model's performance.

**Effect of Positional Coordinates.** In addition to incorporating visual representation of object-centric region proposals into the model, we also adopt the position information (*i.e.*, positional coordinates). The results of whether using positional coordinates are reported in Table 4. Introducing regional coordinates can improve the performance by **0.8%** points.

**Effect of Each Component.** Finally, we showcase the results of using different components of *REVIVE* in Table 6. We can observe that the introduced components can consistently improve the model's performance. Especially for knowledge retrieval, using the regional descriptions can improve the performance of implicit knowledge by **1.2%**, while adopting the regional features can boost the performance of explicit knowledge retrieval by **1.1%**.

The object-centric region features can achieve **1.4%** points improvement, feeding context-aware questions, which can be denoted as prompt "`context: {caption}. question: {question}`", into the answer generative model attain **0.5%** points gain, further introducing regional descriptions (*i.e.*, regional tags) into contexts, *i.e.*, prompt $X$, has **0.7%** points improvement. These results can validate the efficiencies of our proposed components.

Table 6: Ablation study on different components of *REVIVE*. Note that "*Imp.*" and "*R-Imp.*" mean implicit knowledge retrieved without and with the proposed regional descriptions, "*Exp.*" and "*R-Exp.*" mean explicit knowledge retrieved without and with regional features, "*Visual*" represents object-centric region features, "*Context*" and "*Tag*" mean introducing the contexts and regional descriptions (*i.e.* tags) into the final answering model respectively. "*Acc.*" means accuracy.

| Imp. | R-Imp. | Exp. | R-Exp. | Visual | Context | Tag | Acc. (%) |
|---|---|---|---|---|---|---|---|
| ✓ | | | | | | | 51.2 |
| ✓ | ✓ | | | | | | 52.4 |
| ✓ | ✓ | ✓ | | | | | 52.9 |
| ✓ | ✓ | ✓ | ✓ | | | | 54.0 |
| ✓ | ✓ | ✓ | ✓ | ✓ | | | 55.4 |
| ✓ | ✓ | ✓ | ✓ | ✓ | ✓ | | 55.9 |
| ✓ | ✓ | ✓ | ✓ | ✓ | ✓ | ✓ | 56.6 |

## 4.4 Quantitative Result Analysis

Finally, we present the quantitative results and provide analysis for error cases, so that we can have a clear insight into the proposed approach.

**Visualizing Results.** The success cases of our approach are shown in Figure 3. We can observe that our approach can accurately retrieve implicit and explicit knowledge, which corresponds to the detected object regions, and deal with the relationship among these object areas. For example, in

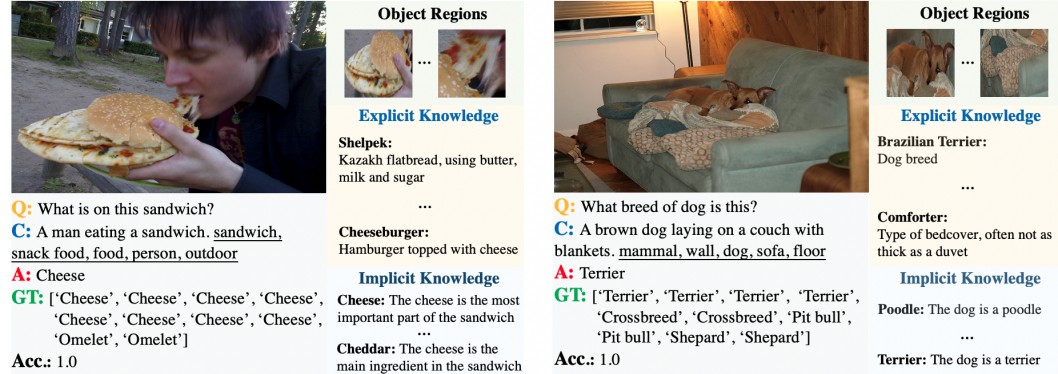

Figure 3: Representative success cases of the proposed *REVIVE* on OK-VQA dataset [22]. *"Q"*, *"C"*, *"A"* and *"GT"* denote question, context, predictive answer, ground truth answers respectively. Note that the underlined text represents regional tags and five tags are selected for illustration. We rescale all the object regions to the same size for a clearer view. *"Acc."* means accuracy.

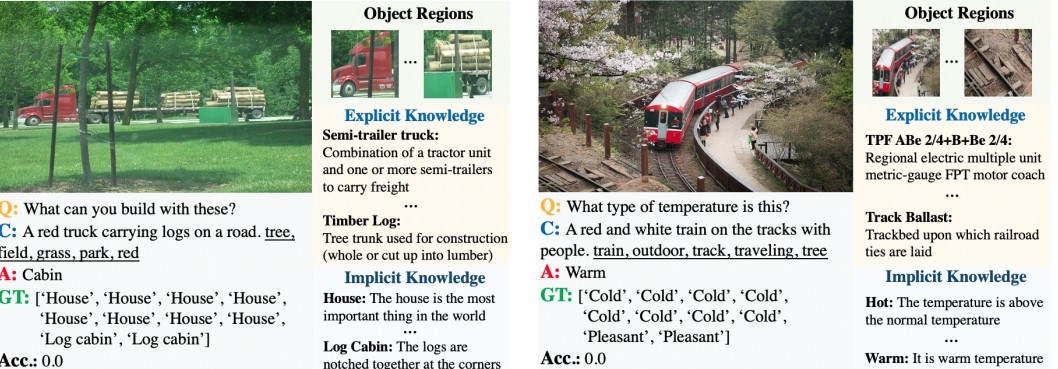

Figure 4: Representative failure cases of the proposed *REVIVE* on OK-VQA dataset [22].

the left example of Figure 3, our method can recognize the potential referred objects and retrieve useful knowledge (*e.g.*, cheeseburger and cheese), thus generating the correct answer, while in the right example, our method can also retrieve important knowledge (*e.g.*, brazilian terrier) to answer the breed of the referred dog.

**Failure Cases Analysis.** We showcase the failure examples in Figure 4. As shown in the left example, even though the prediction result *Cabin* doesn't appear in the ground truth answers, the generated answer of our approach is still reasonable for such scenario. For the right example, our predicted result is wrong due to the difficulty of answering such a general question. From figure 4, we can also observe that our method can generate useful object-centric regions and accurately retrieve corresponding knowledge, especially explicit knowledge, which can demonstrate the potential of the proposed method.

# 5    Limitations and Broader Impact

The quality of constructed Wikidata subset and designed textual prompt can influence final retrieved knowledge. In addition, the detector for obtaining region proposals also affect retrieved knowledge and visual features, all these factors affect the models' performances.

This paper proposes a novel approach *REVIVE* for knowledge-based VQA. *REVIVE* can help models to efficiently use visual and language information sources to answer open-domain questions. It can also generalize to real-life products, *e.g.*, dialogue robot. However, the failure cases of *REVIVE* will be negative to the society when using it as the educational technique. There may also exist certain forms of bias, *i.e.*, the model may predict biased answers if the training data of knowledge-based VQA contain certain bias. For example, [1] suggests the model may be driven by superficial correlations in the training data, and [9] shows the VQA datasets may contain gender and racial bias that may cause the models to learn harmful stereotypes.

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
