# REVIVE: Regional Visual Representation Matters in Knowledge-Based Visual Question Answering (Supplementary Materials)

## A  Overview

In the supplementary materials, we provide the following sections:

(a) Implementation details of implicit knowledge retrieval in Section B.

(b) Ablation study experiments in Section C.

(c) Visualization results in Section D.

## B  Implementation Details of Implicit Knowledge Retrieval

We first describe more implementation details of implicit knowledge retrieval of the proposed *REVIVE*. Specifically, we explain how we extract multiple answer candidates.

**Multiple Candidates.**  We retrieve multiple implicit knowledge candidates for each sample during training and inference stages to improve the robustness of answer generation. Specifically, we follow PICa [6], which proposes to use multi-query ensemble, *i.e.*, they prompt the GPT-3 [1] for $k_1$ times and choose the one with the highest probability as final answer prediction. Compared with PICa's multi-query ensemble approach, we take all these $k_1$ predictions from GPT-3 [1] as implicit knowledge candidates. Note that for each candidate, we also prompt the GPT-3 model to obtain its corresponding explanation. In our experiments, we just retrieve 5 (*i.e.*, $k_1 = 5$) implicit knowledge candidates and corresponding explanations.

## C  Ablation Study

Next, we conduct more ablation study experiments to provide deeper insight into the components of our proposed *REVIVE*.

**The effect of multiple implicit knowledge candidates.**  To validate the influence of the number of retrieved implicit knowledge candidate on the model's performance, we report the results in Table 1. When using only one implicit knowledge candidate, the model can achieve **55.8%** accuracy, after taking 5 implicit knowledge candidates, the performance can be improved to **56.6%** accuracy. However, when the retrieved candidate number is 8, we can see that the performance isn't the best, we conjecture that it's enough to include essential candidates when $k_1 = 5$. Due to certain incorrect answer predictions by GPT-3, larger $k_1$ may introduce incorrect and unnecessary candidates, thus hurting the model's performance by using too much noisy and misleading knowledge.

**The effect of explicit knowledge number.**   Since the number of retrieved explicit knowledge samples can have an effect on the model's performance, we conduct the experiments and show the results in Table 2. We find the model can achieve optimal performance when $k_2 = 40$. It's reasonable to see that a too large $k_2$ (*i.e.*, $k_2 = 50$) cannot let the model achieve optimal performance, since when $k_2$ increases, there will exist certain retrieved explicit knowledge samples which have relatively low confidences, thus introducing unreliable knowledge and hurting the model's performance.

36th Conference on Neural Information Processing Systems (NeurIPS 2022).

Table 1: Ablation study on using different implicit knowledge candidates. $k_1$ represents the number of retrieved implicit knowledge candidates.

| $k_1$ | Accuracy (%) |
|---|---|
| 1 | 55.8 |
| 3 | 56.3 |
| 5 | **56.6** |
| 8 | 56.4 |

Table 2: Ablation study on using different explicit knowledge numbers. $k_2$ represents the number of retrieved explicit knowledge samples.

| $k_2$ | Accuracy (%) |
|---|---|
| 10 | 55.6 |
| 20 | 55.9 |
| 30 | 56.2 |
| 40 | **56.6** |
| 50 | 56.3 |

Table 3: Ablation study on using different object detectors. Note that Faster R-CNN (R50) and Faster R-CNN (R101) mean using ResNet-50 [2] and ResNet-101 [2] as backbones.

| Detector | Accuracy (%) |
|---|---|
| Faster R-CNN (R50) | 55.3 |
| Faster R-CNN (R101) | 55.6 |
| GLIP | **56.6** |

**The effect of using different detectors.** To figure out the effect of choosing different object detectors on the final performances, we show the results of using Faster R-CNN [5] and GLIP [3] in Table 3. We can see that Faster R-CNN with ResNet-50 and ResNet-101 as the backbone can achieve **55.3%** and **55.6%** accuracy respectively, and using the GLIP as the object detector can achieve the optimal performance (*i.e.*, **56.6%**). These results demonstrate the accuracy of detecting object regions play an important role in the final performances.

# D  Visualization Results

Finally, we showcase more visualization cases in Figure 1, 2, 3, 4 and 5. In Figure 1, using the proposed regional descriptions/tags, we can retrieve more accurate implicit knowledge. Taking the top example of Figure 1 for explanation, without introducing the informative regional descriptions (e.g., "sunlight" and "sun"), we cannot generate the correct implicit knowledge candidate "Sun", since the "Lamp" is also reasonable when given the question and context, which can demonstrate the effectiveness of using the regional descriptions for implicit knowledge retrieval.

In Figure 3, 4 and 5, we can see that our proposed method can focus on important object-centric areas, and then retrieve relevant knowledge for corresponding regional areas, which can be used to generate accurate answers. These visualization results can demonstrate the effectiveness and potential of the proposed *REVIVE*.

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

| Image | Question | Context | Regional Tags | Imp. | R-Imp. | GT Answers |
|---|---|---|---|---|---|---|
| | What is the very bright light called above the girl's head in this photo? | A young girl sitting in a car looking at her cell phone. | sunlight, sunshine, car, sky, person, outdoor, clothing, window, mirror, holding | **Lamp:** The lamp can produce bright light | **Sun:** The sun is the brightest object in the sky. | ['Sun', 'Sun', 'Sun', 'Sun', 'Sun', 'Sun', 'Sun', 'Sun', 'Glare', 'Glare'] |
| | Is the bird in the picture a carnivore or herbivore? | A yellow bird is eating from a bird feeder. | bird, finch, animal, canary, beak, bird feeder, sitting, bird food, outdoor, perched | **Carnivore:** The bird eats fish | **Herbivore:** It has a beak, which is used to eat plants. | ['Herbivore', 'Herbivore', 'Herbivore', 'Herbivore', 'Herbivore', 'Herbivore', 'Herbivore', 'Herbivore', 'Herbivore', 'Herbivore'] |
| | What is the name of the cooking style used to produce the items on the table in this photo? | A couple of women standing at a table with food. | shortcake, cake, bread, person, clothing, snack food, dessert, outdoor, group | **Barbecue:** The food is cooked over a fire | **Bake:** The items are cooked in an oven. | ['Bake', 'Bake', 'Bake', 'Bake', 'Bake', 'Bake', 'Bake', 'Bake', 'Stir fry', 'Stir fry'] |

Figure 1: The implicit knowledge retrieval visualization results without and with the proposed regional descriptions/tags. Note that *"Imp."* and *"R-Imp."* mean the implicit knowledge retrieved without and with the regional descriptions/tags. "Regional Tags" represents the proposed regional descriptions. "Context" means the caption. We only use 10 regional tags for illustration.

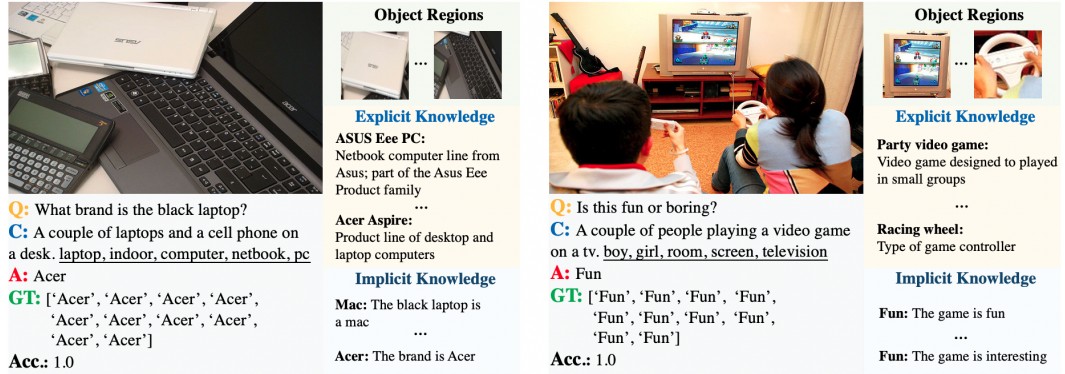

Figure 2: Representative visualization cases of the proposed *REVIVE* on OK-VQA dataset [4]. *"Q"*, *"C"*, *"A"* and *"GT"* denote question, context, predictive answer, ground truth answers respectively. Note that the underlined text represents regional tags and five tags are selected for illustration. We rescale all the object regions to the same size for a clearer view. *"Acc."* means accuracy.

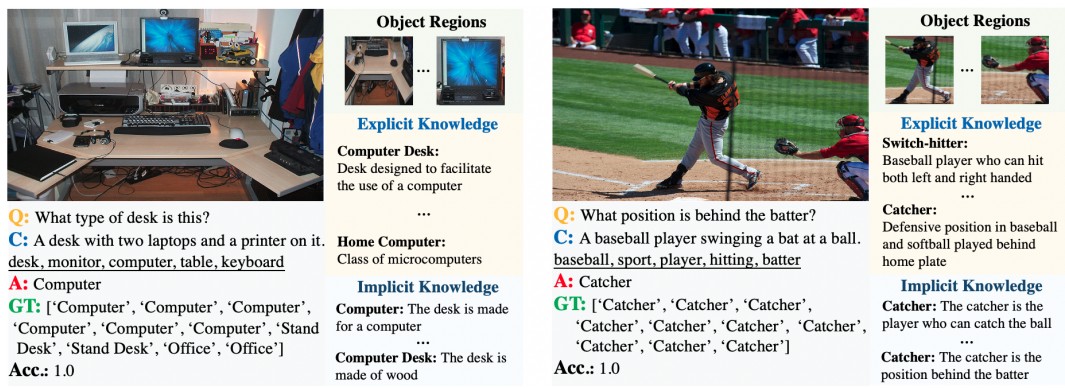

Figure 3: Representative visualization cases of the proposed *REVIVE* on OK-VQA dataset [4]. *"Q"*, *"C"*, *"A"* and *"GT"* denote question, context, predictive answer, ground truth answers respectively. Note that the underlined text represents regional tags and five tags are selected for illustration. We rescale all the object regions to the same size for a clearer view. *"Acc."* means accuracy.

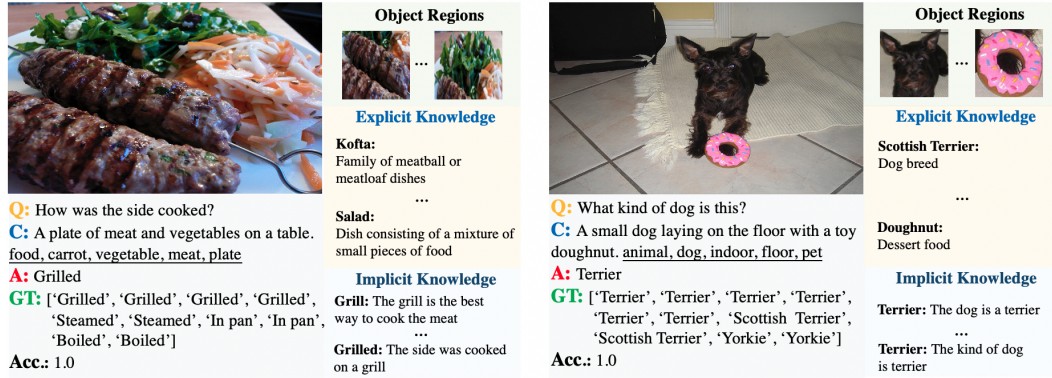

Figure 4: Representative visualization cases of the proposed *REVIVE* on OK-VQA dataset [4]. *"Q"*, *"C"*, *"A"* and *"GT"* denote question, context, predictive answer, ground truth answers respectively. Note that the underlined text represents regional tags and five tags are selected for illustration. We rescale all the object regions to the same size for a clearer view. *"Acc."* means accuracy.

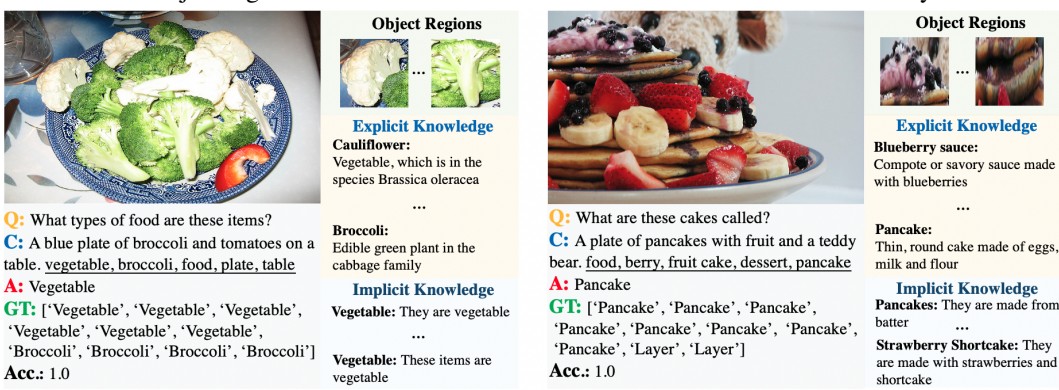

Figure 5: Representative visualization cases of the proposed *REVIVE* on OK-VQA dataset [4]. *"Q"*, *"C"*, *"A"* and *"GT"* denote question, context, predictive answer, ground truth answers respectively. Note that the underlined text represents regional tags and five tags are selected for illustration. We rescale all the object regions to the same size for a clearer view. *"Acc."* means accuracy.

[5] Shaoqing Ren, Kaiming He, Ross Girshick, and Jian Sun. Faster r-cnn: Towards real-time object detection with region proposal networks. *Advances in neural information processing systems*, 28, 2015.

[6] Zhengyuan Yang, Zhe Gan, Jianfeng Wang, Xiaowei Hu, Yumao Lu, Zicheng Liu, and Lijuan Wang. An empirical study of gpt-3 for few-shot knowledge-based vqa. *arXiv preprint arXiv:2109.05014*, 2021.