# OpenReview forum: "REVIVE: Regional Visual Representation Matters in Knowledge-Based Visual Question Answering"
_NeurIPS.cc/2022/Conference — NeurIPS 2022 Accept_

### Official Review · Reviewer_3J5a · 2022-07-06

**Rating:** 6
**Confidence:** 5
**Soundness:** 3 good
**Presentation:** 3 good
**Contribution:** 3 good

**Summary:**

This paper augments the KAT model with regional visual representations for Outside-knowledge Visual Question Answering (OK-VQA) and achieves new state-of-the-art results. Precisely, the regional tags are used for retrieving better implicit knowledge using GPT-3. Regional features are used to retrieve explicit features from WikiData and serve as an additional hint to the FiD answer generator.

**Questions:**

See weakness.  Also, the content of the explicit knowledge may not be relevant to the question, it is also good to do the ablation that only encodes the entities from CLIP without the knowledge sentences to prove the contribution of the regional description.

**Limitations:**

The authors have adequately addressed the limitations and potential negative social impact of their work.

**Strengths And Weaknesses:**

Strengths:
This paper presents a critical weakness of the existing SOTA approach that the regional features are missing. The authors address this issue by introducing the regional descriptions in different formats for different stages of the REVIVE framework including implicit knowledge retrieval and the answer generation stage. The experiments verify the value of introducing the regional features.

Weakness:
The experiement part is not clear in that we do not understand how the regional features benefit the performance. Concretely, as the regional descriptions are injected into different stages of the framework, how do these descriptions influence the performance of implicit knowledge retrieval? Also, the left qualitative example in Figure 3 confuses me as none of the knowledge retrieved mentions the right answer ``battery'' and the regional tags introduce additional misleading objects say the desktop.


Originality: The paper is original.
Quality: The paper presents an interesting idea of using regional features to augment the prompts for better OK-VQA results.
Clarity: The paper is well written.
Significance: Ok,

---

> ### Author Response · Authors · 2022-08-02
> **Response to Reviewer 3J5a (Part 2)**
>
> ### Q3: The content of the explicit knowledge may not be relevant to the question, it is also good to do the ablation that only encodes the entities from CLIP without the knowledge sentences to prove the contribution of the regional description.
>
> **Response:** We appreciate this constructive suggestion, we have performed the ablation study experiment that only encodes the entities from CLIP [2] without the knowledge sentences from the explicit knowledge. The results are shown in the following table, we only use the implicit knowledge with or without the regional descriptions for final answering model in this ablation study.
>
> | Regional Descriptions | Accuracy (%) |
> | :-: | :-: |
> | ✗ | 51.2 |
> | ✓ | **52.4** |
>
> It has been shown that when further introducing the regional descriptions into the textual prompt for implicit knowledge retrieval, the final performance can be improved from **51.2%** to **52.4%** accuracy, *i.e.*, **1.2%** accuracy improvement. The results can prove that using the regional descriptions can more accurately retrieve implicit knowledge, this is reasonable since the regional descriptions can provide more object-centric textual clues for GPT-3 model [1].
>
> We've already added this ablation study experiment into our latest submission (*i.e.*, Table 6).
>
>
> [1] Brown et al. Language models are few-shot learners. NeurIPS 2020
>
> [2] Radford et al. Learning Transferable Visual Models From Natural Language Supervision. ICML 2021
>
> -----
>
> Thanks again for your supportive comments. We hope that our explanations have successfully cleared your concerns

---

> > ### Comment · Reviewer_3J5a · 2022-08-06
> > **response**
> >
> > I am confused about this experiment setting, can you explain more on this?
> > Are the results corresponding to some lines in table 6?
> > What do you mean by "use the implicit knowledge with or without the regional descriptions for final answering model in this ablation study"?
> >
> > REVIVE with only implicit knowledge achieves 52.4 based on the table 6, is it right?

---

> > > ### Author Response · Authors · 2022-08-07
> > > **Response to Reviewer 3J5a**
> > >
> > > Our implicit knowledge is different from what is originally proposed in the KAT paper [1] – we add regional descriptions in the prompts to better retrieve the knowledge that corresponds to the regional concepts. In our original paper, we showed the result **52.4%** with title *implicit knowledge*, which is actually the result of our regional version of implicit knowledge retrieval (and this is implicit knowledge only, i.e., no other modules added).
> > >
> > > Yet the score **52.4%** does not show how the regional descriptions in the implicit knowledge affect the final performance. Therefore, we remove the regional descriptions in the implicit knowledge retrieval, and do retrieval in an identical way to what is proposed in KAT. This leads to the score **51.2%**.
> > >
> > > In the updated paper, we modified Table 6 to include the new ablation study. The first two lines, i.e., *Imp.* and *R-Imp.* corresponds to the two experiments, suggesting adopting regional descriptions in the implicit knowledge can lead to **1.2%** improvement. We hope this answers your question. If you have any further questions or concerns, please do not hesitate to let us know.
> > >
> > > Thank you for your time!
> > >
> > > [1] Liangke Gui, Borui Wang, Qiuyuan Huang, Alex Hauptmann, Yonatan Bisk, and Jianfeng Gao. Kat: A knowledge augmented transformer for vision-and-language

---

> > > > ### Author Response · Authors · 2022-08-08
> > > > **Any more concerns or suggestions?**
> > > >
> > > > Dear Reviewer 3J5a, we really appreciate for your time and efforts in improving our work. Your concerns and suggestions are very valuable for improving the rigorousness, clarity and readability of our paper. We're so pleased to receive such helpful suggestions and we've revised our paper by them. Since the deadline of reviewer-authors discussion is approaching, if you have any concerns or suggestions about our work, please let we know. We are so happy to address your concerns and revise our paper accordingly.

---

> ### Author Response · Authors · 2022-08-02
> **Response to Reviewer 3J5a (Part 1)**
>
> Thanks for appreciating our motivation and method! And also thanks for your constructive feedback and suggestions for our work, we've revised the manuscript to improve its clarity and reader friendliness. The following are our answers to specific questions:
>
> -----
> ### Q1: The experiment part is not clear in that we do not understand how the regional features benefit the performance. Concretely, as the regional descriptions are injected into different stages of the framework, how do these descriptions influence the performance of implicit knowledge retrieval?
>
> **Response:** Thanks for the great suggestion! We have revised our experiment section, especially Table 6, to better highlight how the regional features benefit the performance for different components.
>
> Specifically, for implicit knowledge retrieval, we compare the case with and without regional descriptions, and show that adding the regional descriptions can improve the performance by **1.2%**. Similarly, for explicit knowledge, we consider retrieving it with and without regional features, and show that adopting the regional features can boost the performance by **1.1%**.
>
> Furthermore, the object-centric region features can achieve **1.4%** points improvement, feeding context-aware questions into the answer generative model attain **0.5%** points gain, further introducing regional descriptions (*i.e.*., regional tags) into contexts has **0.7%** points improvement. These results can explain how the regional features in each component benefit the performance.
>
> In order to illustrate how regional information affects the knowledge retrieval process, we add the Figure 1 in the latest submitted supplementary materials due to the space limit of main text. Taking the top example of Figure 1 for explanation, without introducing the informative regional descriptions (e.g., *"sunlight"* and *"sun"*), we cannot generate the correct implicit knowledge candidate *"Sun"*, since the *"Lamp"* is also reasonable when given the question and context, which can demonstrate the effectiveness of using the regional descriptions for implicit knowledge retrieval.
>
> To better figure out the influence of using regional descriptions for implicit knowledge retrieval, we also conduct the ablation study and you can refer it in Q3.
>
> -----
> ### Q2: The left qualitative example in Figure 3 confuses me as none of the knowledge retrieved mentions the right answer "battery'' and the regional tags introduce additional misleading objects say the desktop.
>
> **Response:** Sorry for the confusion. We retrieve 5 implicit knowledge candidates, while we only show the first retrieved implicit knowledge candidate in Figure 3 and Figure 4. The correct answer *"battery"* is also retrieved as the candidate in the implicit knowledge, but it has been omitted by the ellipsis in the left example of Figure 3 due to the limit of space.
>
> The goal of the regional descriptions is to provide a detailed and comprehensive description of the image. Therefore, it inevitably includes irrelevant concepts. Large language models with strong QA capability can naturally pick the relevant concepts from the contexts.
>
> Since this example is not representative enough, we replaced it by another example in the left example of Figure 3. In the new example, the implicit knowledge retrieval is based on the regional description/tag *"sandwich''*, and the correct answer *"cheese''* is successfully retrieved by GPT-3 model [1].
>
> [1] Brown et al. Language models are few-shot learners. NeurIPS 2020

---

### Official Review · Reviewer_5EMT · 2022-07-11

**Rating:** 6
**Confidence:** 4
**Soundness:** 4 excellent
**Presentation:** 4 excellent
**Contribution:** 3 good

**Summary:**

This paper addresses the task of knowledge-based visual question answering (VQA). Given a question and an image, the aim is to answer by using external knowledge bases. The proposed model is built on top of large pre-trained models (GLIP, CLIP, GPT3, Vinyl, etc.) to extract and encode visual features, generate captions, retrieve external knowledge, encode implicit knowledge and predict an answer. The main contribution of the paper is to make better use of the visual information than in previous work, showing in the experiments that visual features can contribute to improving performance on the OK-VQA dataset.

**Questions:**

- Did you compare GLIP and Faster R-CNN object detectors and how they affect the final performance?
- Will the code and models be made publicly available?

**Limitations:**

A more in-depth discussion about bias (both distributional [6] and societal [7]) in knowledge-based VQA and how it affects the proposed model would be interesting, but not necessary.
- [6] Agrawal et al. Don't just assume; look and answer: Overcoming priors for visual question answering. CVPR 2018.
- [7] Hirota et al. Gender and Racial Bias in Visual Question Answering Datasets. ACM FAccT 2022.

**Strengths And Weaknesses:**

Strengths

- The paper and the proposed model are well-motivated. Previous work on knowledge-based VQA did not make full use of the image signal for answering prediction, relying mostly on external knowledge.

- Showing that the information contained in the image is important for answering knowledge-based visual questions is an important contribution to the field. Until now, image features were dismissed and most of the attention was put on the language and knowledge signal. The paper shows that more attention should be paid to visual information.

- The proposed method outperforms previous work on the OK-VQA dataset by a large margin, showing the efficacy of incorporating image information in the answer prediction.

- Ablation study is conducted showing results when different parameters of the model are modified. According to the results, all of the components of the proposed model have a positive contribution to the overall performance.

Weaknesses

- Although previous work did not make full use of the image information on knowledge-based VQA, there are previously proposed methods that incorporate the information in the prediction module (as opposed to L36 “only use visual information for knowledge retrieval but ignore it in the final answering model”). A few examples: [1][2][3] (and others). Also, some work on video knowledge-based VQA also uses captions and object relationships in the prediction module [4][5]. I don’t think this changes the main point of the paper, as these models could not make the visual features improve significantly the final performance, but it would be better to accurately discuss it in the paper.

- Text in the figures looks blurred. The resolution should be improved.

References
- [1] Narasimhan et al. Out of the Box: Reasoning with Graph Convolution Nets for Factual Visual Question Answering. NeurIPS 2018
- [2] Narasimhan et al. Straight to the facts: Learning knowledge base retrieval for factual visual question answering. ECCV 2018
- [3] Shah et al. KVQA: Knowledge-Aware Visual Question Answering. AAAI 2019
- [4] Garcia et al. KnowIT VQA: Answering Knowledge-Based Questions about Videos. AAAI 2020.
- [5] Garcia et al. Knowledge-Based Video Question Answering with Unsupervised Scene Descriptions. ECCV 2020

---

> ### Author Response · Authors · 2022-08-02
> **Response to Reviewer 5EMT**
>
> Thanks for appreciating our motivation and method! And also thanks for your constructive feedback and suggestions for our work, we've revised the manuscript to improve its clarity and reader friendliness. The following are our answers to specific questions:
>
> -----
> ### Q1: Although previous work did not make full use of the image information on knowledge-based VQA, there are previously proposed methods that incorporate the information in the prediction module (as opposed to L36 “only use visual information for knowledge retrieval but ignore it in the final answering model”). ...... but it would be better to accurately discuss it in the paper.
>
> **Response:** Thank you for the great suggestion! We have revised the language in L36 to be more accurate, and also add separate discussions in Section 2 (related works), *i.e.* L76-L78.
>
> In fact, in the previous version of our paper, the L36 *"only use visual information for knowledge retrieval but ignores it in the final answer model"* especially refers to recent SOTA works (*i.e.*, PICa [6] and KAT [7]) for knowledge-based VQA task, they ignore the visual information in the final answering model.
>
> Even though the aforementioned works [1][2][3][4][5] use the visual embeddings or captions in the final prediction module, they haven't used the regional representations to retrieve different types of knowledge and incorporated the object-centric representations into the final prediction model, and they are not directly applicable to improve the encoder-decoder module in knowledge-based VQA task.
>
>
> [1] Narasimhan et al. Out of the Box: Reasoning with Graph Convolution Nets for Factual Visual Question Answering. NeurIPS 2018
>
> [2] Narasimhan et al. Straight to the facts: Learning knowledge base retrieval for factual visual question answering. ECCV 2018
>
> [3] Shah et al. KVQA: Knowledge-Aware Visual Question Answering. AAAI 2019
>
> [4] Garcia et al. KnowIT VQA: Answering Knowledge-Based Questions about Videos. AAAI 2020.
>
> [5] Garcia et al. Knowledge-Based Video Question Answering with Unsupervised Scene Descriptions. ECCV 2020
>
> [6] Yang et al. An empirical study of GPT-3 for few-shot knowledge-based VQA. AAAI 2021
>
> [7] Gui et al. KAT: A knowledge augmented transformer for vision-and-language. NAACL 2022
>
> -----
>
> ### Q2: Text in the figures looks blurred. The resolution should be improved.
>
> **Response:** Thank you for pointing this out! We have updated all the figures in our paper so that the texts in the figures are clearer.
>
> -----
> ### Q3: Did you compare GLIP and Faster R-CNN object detectors and how they affect the final performance?
>
> **Response:** Yes, we have performed this experiment, so that we can better figure out the influence of using different object detectors on the model's final performance. The results of using the GLIP [1] and Faster R-CNN [2] as the object detectors are reported in the following table.
>
> | Detector | Accuracy (%) |
> | :-: | :-: |
> | Faster R-CNN (R50)| 55.3 |
> | Faster R-CNN (R101) | 55.6 |
> | GLIP | **56.6** |
>
> As shown in the table, we can see that Faster R-CNN with ResNet-50 [3] and ResNet-101 [3] as the backbones can achieve **55.3%** and **55.6%** accuracy, respectively, and using the GLIP as the object detector can achieve better performance (*i.e.*, **56.6%**). These results demonstrate the accuracy of detecting object regions can play an important role in the final performance.
>
> Due to the limit of space, we added this ablation study experiment into our latest submission of supplementary materials (*i.e.*, L34-L39 and Table 3).
>
> [1] Li et al. GLIP: Grounded Language-Image Pre-training. CVPR 2022
>
> [2] Ren et al. Faster R-CNN: Towards real-time object detection with region proposal networks. NeurIPS 2015
>
> [3] He et al. Deep residual learning for image recognition. CVPR 2016
>
> -----
>
> ### Q4: Will be the codes and models released?
>
> **Response:** Yes, we will make the code and models publicly available upon the acceptance of the paper.
>
> -----
>
> ### Q5: A more in-depth discussion about bias (both distributional and societal) in knowledge-based VQA and how it affects the proposed model would be interesting.
>
> **Response:** We appreciate this helpful suggestion. We have revised the limitation and made a more in-depth discussion about bias accordingly in our latest submission (*i.e.*, L311-L315).
>
> -----
>
> Thanks again for your supportive comments. We hope that our explanations have successfully cleared your concerns

---

> > ### Author Response · Authors · 2022-08-08
> > **Any more concerns or suggestions?**
> >
> > Dear Reviewer 5EMT, we would like to thank you again for your precious efforts, time and suggestions. Your constructive suggestions (e.g., figure, ablation study, wording and etc.) have significantly improved the quality of our paper. Considering the deadline of reviewer-authors discussion is approaching, if you have any concerns or suggestions about our work, please let we know. We are happy to address your concerns and revise our paper accordingly.

---

> > > ### Comment · Reviewer_5EMT · 2022-08-09
> > > **Response**
> > >
> > > Thank you for the detailed author response. After reading all the reviews and the discussion, I am happy to support this paper for acceptance.

---

> > > > ### Author Response · Authors · 2022-08-09
> > > > **Thanks for your feedback!**
> > > >
> > > > Dear Reviewer 5EMT, thanks for your effort again! We are happy that our rebuttal well addressed your concerns!

---

### Official Review · Reviewer_uNM2 · 2022-07-13

**Rating:** 5
**Confidence:** 4
**Soundness:** 2 fair
**Presentation:** 2 fair
**Contribution:** 1 poor

**Summary:**

The authors observe that in most state-of-the-art knowledge-based VQA methods:
1) visual features are extracted either from the whole image or in a sliding window manner for retrieving knowledge, and the important relationship within/among object regions is neglected;
2) visual features are not well utilized in the final answering model, which is counter-intuitive to some extent.
Based on these observations, they propose a new knowledge-based VQA method REVIVE, which tries to utilize the explicit information of object regions not only in the knowledge retrieval stage but also in the answering model.
The authors perform several experiments on the standard OK-VQA dataset and achieve new state-of-the-art performance.


**Questions:**

It would be better to further emphasize the necessity of each of the proposed components in the methodology section. What are the technical reasons why each of the components is required for better training and performance improvement?

**Limitations:**

The authors addressed the limitations of the proposed method and the potential negative social impact of their work.

**Strengths And Weaknesses:**

Strength

- The proposed method shows favorable performance compared to the recent baselines.

- The authors provide various analyses on the design choices including the hyper-parameters that affect the performance improvements.

Weakness

- The technical novelty in this paper is too weak. This paper is an engineering paper that combines several engineering tricks without a unified motivation or a theoretical background. Most of the components of the proposed method are hackish tricks such as changing the input cues of some modules in their model.

- In the ablation study in Table 6, the most effective components that give the largest performance gain are explicit knowledge (which is already introduced in the KAT method) and ensembling (which is a well-known trick to boost the performance). The other components such as changing or adding input cues show almost negligible performance gain which is less than 1%.



========== ------- Comments after the rebuttal ------========
I have read the other reviewer's comments and the author's rebuttal, which addresses most of my concerns. Therefore, I would like to raise my score.

---

> ### Author Response · Authors · 2022-08-02
> **Response to Reviewer uNM2 (Part 2)**
>
> ### Q3: It would be better to further emphasize the necessity of each of the proposed components in the methodology section. What are the technical reasons why each of the components is required for better training and performance improvement?
>
> **Response:** Thanks for the great suggestion! We have revised our methodology section to highlight our contribution in systematically incorporating visual signals.
>
> We summarize the motivation of each component as follows.
>
>  (a) **Implicit knowledge with regional descriptions.** GPT-3 [1] is a powerful language model with question-answering capability. Yet it only accepts language input. So we convert the question-image pairs into textual formats. In addition to questions and captions, we further introduce the regional descriptions, which can provide more regional information. The ablation study on introducing the regional descriptions/tags into the prompt is in the first table above and Q3 of reviewer 3J5a.
>
> (b) **Explicit knowledge with regional features.**  KAT [2] uses a sliding window on the image to retrieve explicit knowledge, which hurt the performance by unavoidably introducing much irrelevant background information. Instead, we propose to use the regional features obtained by an object detector to retrieve explicit knowledge. The ablation study on using the explicit knowledge retrieved by our proposed region-based manner against KAT is shown in the first table above.
>
> (c) **Object-centric Representations.** Instead of only using language clues like PICa [3] and KAT [2], we further integrate the visual information into the final answering model, and we find the positional information of the objects matters, thus we encode the regional features and the positional coordinates with a visual encoder as the object-centric representation, the ablation can also be referred to the first table above.
>
> The performance improvement of each proposed component can be observed in the table in Q1 and Table 6 in our latest submitted paper.
>
> [1] Brown et al. Language models are few-shot learners. NeurIPS 2020
>
> [2] Gui et al. KAT: A knowledge augmented transformer for vision-and-language. NAACL 2022
>
> [3] Yang et al. An empirical study of GPT-3 for few-shot knowledge-based VQA. AAAI 2021
>
>
>
> ----
>
>
> Thanks again for your supportive comments. We hope that our explanations have successfully cleared your concerns

---

> > ### Author Response · Authors · 2022-08-03
> > **Any more concerns or suggestions?**
> >
> > Dear Reviewer uNM2, we would like to thank you again for the efforts and suggestions. We have provided the detailed responses to all your concerns. Could you take a look at our response? Feel free to raise any more questions, we are happy to answer them further.

---

> > > ### Author Response · Authors · 2022-08-08
> > > **We are happy to address any remaining concerns or questions**
> > >
> > > Dear Reviewer uNM2, the deadline of reviewer-authors discussion is Aug 09 '22 01:00 PM PDT, which means the time left for discussion is only about one day. If you have any further concerns or questions, could you please be explicit about it? We will try our best to address it.
> > >
> > > We really appreciate your help in improving our work!

---

> > ### Comment · Reviewer_uNM2 · 2022-08-08
> > **Response**
> >
> > Thank you, authors, for your elaborate rebuttal. I have read the other reviewer's comments and the author's rebuttal, which addresses most of my concerns. Therefore, I would like to raise my score to 5.

---

> > > ### Author Response · Authors · 2022-08-09
> > > **Thanks for the feedback!**
> > >
> > > Dear reviewer uNM2, thanks for your effort again! We are happy that our rebuttal well addressed your concerns!

---

> ### Author Response · Authors · 2022-08-02
> **Response to Reviewer uNM2 (Part 1)**
>
> Thank you for acknowledging the strong performance and extensive experiments of our work! And also thanks for your valuable feedback! We've revised the manuscript to improve its clarity and reader friendliness. The following are our answers to specific questions:
>
> ----
> ### Q1: The technical novelty in this paper is too weak ...... hackish tricks such as changing the input cues of some modules in their model.
>
> **Response:** Our unified motivation is to make full use of the visual features (*i.e.*, regional features) in knowledge-based VQA tasks, which are neglected by existing works. Specifically, we use visual features in both knowledge retrieval and answer prediction:
>
> + Local visual features are important in retrieving external knowledge, as the retrieved knowledge should also correspond to individual concepts in the images, in addition to the global semantics. Therefore, we use extracted regional features to retrieve external knowledge and regional descriptions to obtain implicit knowledge.
> + The final prediction model answers the question based on the retrieved knowledge, which should be given the opportunity to look at the image thoroughly. Therefore, we extend the language encoder-decoder model, *i.e.*, T5 [1], to incorporate the regional features and region coordinates.
>
>
>
> Regional descriptions incorporated in each step above can improve the scores significantly and are critical for achieving state-of-art performance, as shown in the table below. One component is removed for each score.
>
> | Model | Accuracy (%) |
> |:-| :-: |
> | REVIVE | 56.6  |
> | Replace regional features with sliding window features [2] in explicit knowledge retrieval | 55.8 |
> | Remove regional descriptions in implicit knowledge retrieval   |55.6  |
> | Remove object-centric region features in final answering model  |  55.0 |
> | Remove regional descriptions in final answering model  | 55.9 |
>
> Building upon the implicit knowledge module and explicit module from previous works, our work successfully fills in the gap between the existing methods and the value of precise visual modeling.
>
> We have revised the methodology section to highlight our contributions.
>
> [1] Raffel et al. Exploring the Limits of Transfer Learning with a Unified Text-to-Text Transformer. JMLR 2020
>
> [2] Gui et al. KAT: A knowledge augmented transformer for vision-and-language. NAACL 2022
>
> ----
>
> ### Q2: In the ablation study in Table 6, the most effective components that give the largest performance gain are explicit knowledge (which is already introduced in the KAT method) and ensembling (which is a well-known trick to boost the performance). The other components such as changing or adding input cues show almost negligible performance gain which is less than 1%.
>
> **Response:** This is not true. As shown in Table 6, using the visual representations of object regions can improve the performance from **54.0%** to **55.4%** (**+1.4%**).
>
> Furthermore, the implicit/explicit knowledge module in Table 6 is different from what is proposed in KAT [1], we adopt our proposed regional descriptions and regional features for implicit and explicit knowledge retrieval. To illustrate this, we performed a further ablation experiment comparing the performances of using KAT's and the proposed REVIVE's method for retrieving explicit knowledge, which can be referred to the above table in Q1.
>
>
> We have also modified Table 6 to make it clearer. Using the regional descriptions can improve the performance of implicit knowledge by **1.2%**, while adopting the regional features can boost the performance of explicit knowledge retrieval by **1.1%**.
>
> [1] Gui et al. KAT: A knowledge augmented transformer for vision-and-language. NAACL 2022

---

### Author Response · Authors · 2022-08-02
**Summary of changes**

We sincerely thank all the reviewers for their previous time and efforts in reading and reviewing our paper. We are so glad that all reviewers recognize our strong performance and extensive experiments. Your great suggestions have greatly improved the quality of our paper. Please kindly refer to the individual responses below for our response to each question. We have also updated our submission to reflect reviewers’ opinions in details.

+ Section 4.3 and Table 6: We update the ablation study to better show how each of our proposed component improve the performance. In Table 6, we clearly show the performance improvement from implicit knowledge retrieval with the proposed regional descriptions.

+ Section 5: We update the limitations and broader impact, in which we make a more in-depth discussion about bias (both distributional [1] and societal [2]) in knowledge-based VQA.

+ We update all the figures so that their texts are clear, and replace the left example in Figure 3 with a more representative one.

+ Supplementary Materials Section D: We add more visualized VQA examples (*i.e.*, Figure 1, 4 and 5). Especially, in Figure 1 we illustrate the difference of the retrieved implicit knowledge  without/with the proposed regional descriptions.

+ Supplementary Materials Section C: We add a new ablation study on the effect of using different detectors to compare the performances by using different detectors (*e.g.*, Faster R-CNN [3] and GLIP [4]).

+ L36: We adjust the wording to make it clearer.

+ Section 2: We include more discussions on  works that also incorporate visual embeddings and captions.

+ Section 3.4: We revise the relationship to existing works to better clarify the difference between our method and existing methods on knowledge-based VQA tasks.


[1] Agrawal et al. Don't just assume; look and answer: Overcoming priors for visual question answering. CVPR 2018.

[2] Hirota et al. Gender and Racial Bias in Visual Question Answering Datasets. ACM FAccT 2022.

[3] Ren et al. Faster R-CNN: Towards real-time object detection with region proposal networks. NeurIPS 2015

[4] Li et al. GLIP: Grounded Language-Image Pre-training. CVPR 2022

---

### Meta-Review · Area_Chair_71Ad · 2022-08-23

**Recommendation:** Accept
**Confidence:** Certain

**Metareview:**

The paper incorporates regional features to better retrieve relevant knowledge and makes direct use of the visual signal in answer prediction whereas the previous SOTA methods simply rely on the retrieved knowledge for the final prediction. The proposed method outperforms SOTA on OK-VQA by a large margin effectively showing the efficacy of the direct use of visual information in the answer prediction. I agree with the reviewer 5EMT that showing that the information contained in the image is important for answering knowledge-based visual questions is an important contribution to the field as most of the attention was put on the language and knowledge signal.

The author rebuttal also resolved most of the reviewers’ concerns and questions, and made the reviewers reach a consensus towards the acceptance.

**Award:**

No

---

### Decision · Program_Chairs · 2022-09-14

Accept